# REVISITING THE ENTROPY SEMIRING FOR NEURAL SPEECH RECOGNITION

**Oscar Chang, Dongseong Hwang, Olivier Siohan**
Google
{oscarchang,dongseong,siohan}@google.com

## ABSTRACT

In streaming settings, speech recognition models have to map sub-sequences of speech to text before the full audio stream becomes available. However, since alignment information between speech and text is rarely available during training, models need to learn it in a completely self-supervised way. In practice, the exponential number of possible alignments makes this extremely challenging, with models often learning peaky or sub-optimal alignments. Prima facie, the exponential nature of the alignment space makes it difficult to even quantify the uncertainty of a model's alignment distribution. Fortunately, it has been known for decades that the entropy of a probabilistic finite state transducer can be computed in time linear to the size of the transducer via a dynamic programming reduction based on semirings. In this work, we revisit the entropy semiring for neural speech recognition models, and show how alignment entropy can be used to supervise models through regularization or distillation. We also contribute an open-source implementation of CTC and RNN-T in the semiring framework that includes numerically stable and highly parallel variants of the entropy semiring. Empirically, we observe that the addition of alignment distillation improves the accuracy and latency of an already well-optimized teacher-student distillation model, achieving state-of-the-art performance on the Librispeech dataset in the streaming scenario.

## 1 INTRODUCTION

Modern automatic speech recognition (ASR) systems deploy a single neural network trained in an end-to-end differentiable manner on a paired corpus of speech and text (Graves et al., 2006; Graves, 2012; Chan et al., 2015; Sak et al., 2017; He et al., 2019). For many applications like providing closed captions in online meetings or understanding natural language queries for smart assistants, it is imperative that an ASR model operates in a streaming fashion with low latency. This means that before the full audio stream becomes available, the model has to produce partial recognition outputs that correspond to the already given speech.

Ground truth alignments that annotate sub-sequences of speech with sub-sequences of text are hard to collect, and rarely available in sufficient quantities to be used as training data. Thus, ASR models have to learn alignments from paired examples of un-annotated speech and text in a completely self-supervised way. The two most popular alignment models used for neural speech recognition today are Connectionist Temporal Classification (CTC) (Graves et al., 2006) and Recurrent Neural Network Transducer (RNN-T) (Graves, 2012; He et al., 2019). They formulate a probabilistic model over the alignment space, and are trained with a negative log-likelihood (NLL) criterion.

Despite the widespread use of CTC and RNN-T, ASR models tend to converge to peaky or sub-optimal alignment distributions in practice (Miao et al., 2015; Liu et al., 2018; Yu et al., 2021). Prior work outside of ASR has discovered that the standard NLL loss generally leads to over-confident predictions (Pereyra et al., 2017; Xu et al., 2020). Even within ASR, there is strong theoretical evidence that sub-optimal alignments are an inevitable consequence of the NLL training criterion (Zeyer et al., 2021; Blondel et al., 2021).

A common remedy for mitigating over-confident predictions is to impose an entropy regularizer to encourage diversification. For example, label smoothing leads to more calibrated representations and better predictions (Müller et al., 2019; Meister et al., 2020). Another popular technique is

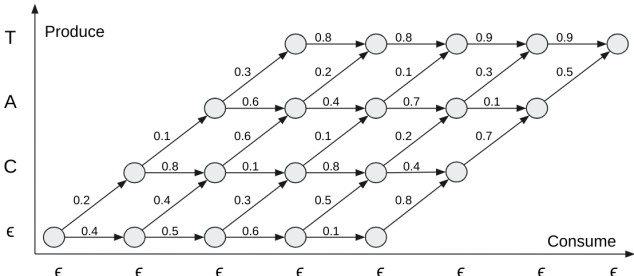

Figure 1: Example of a loop-skewed RNN-T lattice annotated with transition probabilities. In this example, there are 3 text tokens to be produced and 4 acoustic tokens to be consumed, which results in a total of $\frac{(3+4)!}{3!4!} = 35$ alignments and $2*3*4 + 3 + 4 = 31$ transitions. The naive calculation of entropy requires $35*(3+4+1) = 280$ multiplications and $35 - 1 = 34$ additions, while the entropy semiring calculation requires $31*3 = 93$ multiplications and $2*3*4 + 31 = 55$ additions.

knowledge distillation, where we minimize the relative entropy between a teacher's and student's soft predictions, instead of training on hard labels (Hinton et al., 2015; Stanton et al., 2021). However, it is not straightforward to calculate the entropy of the alignment distribution of a neural speech model. This is because the total number of possible alignments is exponential in the length of the acoustic and text sequence, which makes naive calculation intractable.

Fortunately, classical results from Eisner (2001); Cortes et al. (2006) show that the entropy of a probabilistic finite state transducer can be computed in time linear to the size of the transducer. Their approach is based on the semiring framework for dynamic programming (Mohri, 1998), which generalizes classical algorithms like forward-backward, inside-outside, Viterbi, and belief propagation. The unifying algebraic structure of all these classical algorithms is that state transitions and merges can be interpreted as generalized multiplication and addition operations on a semiring. Eisner (2001); Cortes et al. (2006) ingeniously constructed a semiring that corresponds to the computation of entropy.

While these results have been long established, open-source implementations like the OpenFST library (Allauzen et al., 2007) have not been designed with modern automatic differentiation and deep learning libraries in mind. This is why thus far, supervising training with alignment entropy regularization or distillation has not been part of the standard neural speech recognition toolbox. In fact, implementing the entropy semiring on top of modern ASR lattices like CTC or RNN-T is highly non-trivial. During training, we need to compute not only the entropy in the forward pass, but also its gradients in the backward pass, which necessitates a highly numerically stable implementation. Note that even for simple operations like calculating the binary cross entropy of a softmax distribution, naive implementations that do not use the LogSumExp trick are plagued by numerical inaccuracies. When it comes to calculating the entropy of a sequence that might be thousands of tokens long, such inaccuracies accumulate quickly, leading to NaNs that make training impossible. Moreover, it is crucial that the addition of the alignment entropy supervision does not incur additional forward passes through the ASR lattice beyond the one already done to compute NLL.

**Our Contributions** We contribute an open-source implementation of CTC and RNN-T in the semiring framework that is both numerically stable and highly parallel. Regarding numerical stability, we find that the vanilla entropy semiring from Eisner (2001); Cortes et al. (2006) produces unstable outputs not just at the final step, but also during intermediate steps of the dynamic program. Thus, a naive implementation will result in instability in both the forward and backward pass, since automatic differentiation re-uses activations produced during the intermediate steps. To address this, we propose a novel variant of the entropy semiring that is isomorphic to the original, but is designed to be numerically stable for both the forward and backward pass. Regarding parallelism, our implementation allows for efficient plug-and-play computations of arbitrary semirings in the dynamic programming graphs of CTC and RNN-T. Thus, when outputs from more than one semiring are desired, we can compute them in parallel using a single pass over the same data, by simply plugging in a new semiring that is formed via the concatenation of existing semirings.

Beyond our open-source contribution, we also experimentally validate the effectiveness of alignment supervision for regularization and distillation in Section 5. Our first experiment targets small-capacity models that are more likely to learn sub-optimal alignments. In Section 5.1, we show how alignment entropy regularization can reduce the word error rates (WER) of small LSTM models by up to $6.5\%$ and small Conformer models by up to $2.9\%$ relative to the baseline. These results highlight that the optimization objective can significantly impact the performance of the same model. Next in Section 5.2, we propose a novel distillation objective, which we term semiring distillation. Under this objective, the teacher model uses the uncertainty of both the token predictions and the alignments to supervise the student. When applied to a well-optimized $0.6$B parameter RNN-T model, the combined uncertainty produces better WER results in the student than either uncertainty term alone, achieving state-of-the-art results on the Librispeech dataset in the streaming setting. An ablation study further reveals that while distillation in the token space has a small negative effect on latency, distillation in the alignment space reduces emission latency significantly.

The rest of the paper is organized as such: Section 2 presents the semiring framework for dynamic programming. Section 3 provides a brief summary of alignment models for neural speech recognition. Section 4 discusses our implementation of the semiring framework. Section 5 showcases applications for regularization and distillation. Finally, Section 6 concludes the paper. Extra discussions about related work, proofs, and further analysis can be found in the Appendix.

## 2 SEMIRING FRAMEWORK FOR DYNAMIC PROGRAMMING

We begin by introducing definitions for a semiring, and adapting Mohri (1998)'s semiring framework for dynamic programming to weighted directed acyclic graphs (DAG). Unlike Mohri (1998), which applies to single-source directed graphs, we consider DAGs with multiple roots, since they appear more commonly in ASR, for example the CTC lattice.

**Definition 2.1.** A *monoid* is a set $M$ equipped with a binary associative operation $\odot$, i.e. $\forall a, b, c \in M, (a \odot b) \odot c = a \odot (b \odot c)$, and an identity element $e \in M$, i.e. $\forall a \in M, a \odot e = e \odot a = a$. $M$ is called a *commutative monoid* if $\odot$ is commutative, i.e. $\forall a, b \in M, a \odot b = b \odot a$.

**Definition 2.2.** A *semiring* is a set $R$ equipped with addition $\oplus$ and multiplication $\otimes$ such that:

1. $(R, \oplus)$ is a commutative monoid with identity element $\bar{0}$.

2. $(R, \otimes)$ is a monoid with identity element $\bar{1}$.

3. $\otimes$ distributes over $\oplus$ from both the left and right, i.e. $\forall a, b, c \in R$,
   $a \otimes (b \oplus c) = (a \otimes b) \oplus (a \otimes c)$ and $(b \oplus c) \otimes a = (b \otimes a) \oplus (c \otimes a)$.

4. $\bar{0}$ is an annihilator for $(R, \otimes)$, i.e. $\forall a \in R, a \otimes \bar{0} = \bar{0} \otimes a = \bar{0}$.

**Definition 2.3.** Let $G = (V, E)$ be a DAG where every edge $e \in E$ is assigned a weight $w \colon E \to \mathbb{R}$. A *maximal path* of $G$ is a path starting from a root of $G$ to one of $G$'s leaves. The weight $w(\pi)$ of a path $\pi$ is the product of weights of edges in the path. A *computation* of $G$ is the sum of the weights of all the maximal paths in $G$, i.e.

$$C(G, w, \oplus, \otimes) = \bigoplus_{\pi \in \mathit{MaxPaths}(G)} \bigotimes_{e \in \pi} w(e). \tag{1}$$

Even though *MaxPaths*$(G)$ might be exponentially big, because $\otimes$ distributes over $\oplus$ in a semiring, only 1 multiplication operation is needed for each edge, resulting in linear complexity $\mathcal{O}(|V| + |E|)$.

**Theorem 2.4.** *For a DAG $G = (V, E)$, if $(w(E), \oplus, \otimes, \bar{0}, \bar{1})$ is a semiring, then $C(G, w, \oplus, \otimes)$ can be computed in linear time $\mathcal{O}(|V| + |E|)$ via dynamic programming.*

**Proof.** *Sort the vertices in topological order, and enumerate them as $v_1, \ldots, v_n$. Then in that order, inductively accumulate the weights from the edges into the vertices as follows. We slightly abuse notation by over-loading $w$ to also be a function that assigns weights to vertices.*

$$w(v_i) = \begin{cases} \bar{1} & \text{if } v_i \text{ is a root.} \\ \bigoplus_{e=(v,v_i) \in E} w(v) \otimes w(e) & \text{otherwise.} \end{cases} \tag{2}$$

*Let $\Pi_v$ denote the set of paths ending at vertex $v$ that start from a root. We now prove by induction that $w(v_i) = \bigoplus_{\pi \in \Pi_{v_i}} w(\pi)$. The base case of $v_1$ is true by definition given that it is a root. For the inductive step, we assume that $\forall j \leq k, w(v_j) = \bigoplus_{\pi \in \Pi_{v_j}} w(\pi)$. Observe that*

$$
\begin{aligned}
w(v_{k+1}) &= \bigoplus_{e=(v,v_{k+1})\in E} w(v) \otimes w(e) = \bigoplus_{e=(v,v_{k+1})\in E} \left( \bigoplus_{\pi \in \Pi_v} w(\pi) \right) \otimes w(e) \\
&= \bigoplus_{e=(v,v_{k+1})\in E} \left( \bigoplus_{\pi \in \Pi_v} w(\pi) \otimes w(e) \right) = \bigoplus_{\pi \in \Pi_{v_{k+1}}} w(\pi).
\end{aligned}
\tag{3}
$$

*The sum of all the weights in the leaves yields the desired computation in a single pass over $G$.*

Below, we walk the reader through simple examples to demonstrate how Theorem 2.4 works.

**Definition 2.5.** The *probability semiring* can be defined as such: $w(e) = p(e)$ where $p: E \to [0,1]$ is the probability function, $\oplus = +, \otimes = \times, \bar{0} = 0, \bar{1} = 1$.

**Example 2.6.** *Consider the following DAG with the probability semiring.*

$w(v_1) = w(v_2) = 1.$
$w(v_3) = (w(v_1) \otimes w(e_1)) \oplus (w(v_2) \otimes w(e_2)) = p(e_1) + p(e_2).$
$w(v_4) = w(v_3) \otimes w(e_3) = p(e_1)p(e_3) + p(e_2)p(e_3).$
$w(v_5) = w(v_3) \otimes w(e_4) = p(e_1)p(e_4) + p(e_2)p(e_4).$
$C_{p(\pi)} = w(v_4) \oplus w(v_5) = p(e_1)p(e_3) + p(e_2)p(e_3) + p(e_1)p(e_4) + p(e_2)p(e_4).$

In practice, it is common to perform multiplication of probabilities in the log space instead for numerical stability. Under the semiring framework, we can see that this is the same dynamic programming computation but with a different semiring.

**Definition 2.7.** The *log semiring* can be defined as such: $w(e) = \log p(e), a \oplus b = \log(e^a + e^b), a \otimes b = a + b, \bar{0} = -\infty, \bar{1} = 0.$

**Example 2.8.** *Consider the DAG in Example 2.6 with the log semiring. (Details in Appendix C.1)*

$$
C_{\log p(\pi)} = \log \left[ p(e_1)p(e_3) + p(e_2)p(e_3) + p(e_1)p(e_4) + p(e_2)p(e_4) \right].
$$

## 2.1 ENTROPY SEMIRING

Notice that setting $w(e) = p(e) \log p(e)$ with either the probability semiring or the log semiring will not compute the entropy. The salient insight made by Eisner (2001); Cortes et al. (2006) is that entropy can be calculated via a semiring that derives its algebraic structure from the dual number system. Dual numbers are hyper-complex numbers that can be expressed as $a + b\epsilon$ such that $a \in \mathbb{R}, b \in \mathbb{R}, \epsilon^2 = 0$ where addition and multiplication are defined as follows.

$$
\begin{aligned}
\text{Addition:} \quad & (a + b\epsilon) + (c + d\epsilon) = (a + c) + (b + d)\epsilon. \\
\text{Multiplication:} \quad & (a + b\epsilon) \times (c + d\epsilon) = ac + (ad + bc)\epsilon.
\end{aligned}
\tag{4}
$$

Observe that dual numbers form a commutative semiring with the additive identity $0 + 0\epsilon$ and multiplicative identity $1 + 0\epsilon$. Now if we use dual-valued weights $w: E \to \mathbb{R} \times \mathbb{R}$ with the real component representing the probability $p(e)$ and the imaginary component representing the negative entropy $p(e) \log p(e)$, the entropy of a weighted DAG can be computed as before via Theorem 2.4.

**Definition 2.9.** The *entropy semiring* can be defined as such:

$w(e) = \langle p(e), p(e) \log p(e) \rangle,$

$\langle a, b \rangle \oplus \langle c, d \rangle = \langle a + c, b + d \rangle,$

$\langle a, b \rangle \otimes \langle c, d \rangle = \langle ac, ad + bc \rangle,$

$\bar{0} = \langle 0, 0 \rangle, \bar{1} = \langle 1, 0 \rangle.$

**Example 2.10.** *Consider the DAG in Example 2.6 with the entropy semiring. (Details in Appendix C.2)*

$$C_{\langle p(\pi), p(\pi) \log p(\pi) \rangle} = \langle C_{p(\pi)}, \ p(e_1)p(e_3) \log [p(e_1)p(e_3)] + p(e_2)p(e_3) \log [p(e_2)p(e_3)]$$
$$+ p(e_1)p(e_4) \log [p(e_1)p(e_4)] + p(e_2)p(e_4) \log [p(e_2)p(e_4)] \rangle.$$

## 3  NEURAL SPEECH RECOGNITION

Speech recognition is the task of transcribing speech $x$ into text $y$, and can be formulated probabilistically as a discriminative model $p(y|x)$. Because $x$ and $y$ are represented as sequences of vectors, we can define an alignment model that maps sub-sequences of $x$ to sub-sequences of $y$. The alignments allow a speech recognition model to emit a partial sub-sequence of $y$ given a partial sub-sequence of $x$, and thereby work in a streaming fashion. Below, we briefly recap CTC and RNN-T, which are the two most widely used alignment models for neural speech recognition.

Let the acoustic input be denoted as $x = x_{1:T}$ where $x_t \in \mathbb{R}^d$ and the text labels be denoted as $y = y_{1:U}$ where $y_u \in \mathcal{V}$ are vocabulary tokens (we use : to denote an inclusive range). The likelihood of a CTC model is defined as:

$$P(y|x) = \sum_{\hat{y} \in \mathcal{A}_{\text{CTC}}(x,y)} \prod_{t=1}^{T} P(\hat{y}_t | x_{1:t}). \tag{5}$$

where $\hat{y} = \hat{y}_{1:T} \in \mathcal{A}_{\text{CTC}} \subset \{\mathcal{V} \cup \epsilon\}^T$ corresponds to alignments such that removing blanks $\epsilon$ and repeated symbols from $\hat{y}$ results in $y$. CTC makes an assumption of conditional independence, so the likelihoods of every token are independent of each other given the acoustic input.

The likelihood of an RNN-T model is defined as:

$$P(y|x) = \sum_{\hat{y} \in \mathcal{A}_{\text{RNN-T}}(x,y)} \prod_{i=1}^{T+U} P(\hat{y}_i | x_{1:t_i}, y_{1:u_{i-1}}). \tag{6}$$

where $\hat{y} = \hat{y}_{1:T+U} \in \mathcal{A}_{\text{RNN-T}} \subset \{\mathcal{V} \cup \epsilon\}^{T+U}$ corresponds to alignments such that removing blanks $\epsilon$ from $\hat{y}$ results in $y$. Unlike CTC, the likelihood of each token in RNN-T depends on the history of tokens that come before it.

CTC and RNN-T lattices are DAGs where the model likelihoods can be computed efficiently via dynamic programming by transforming the sum-product form of Eqs. (5) and (6) into product-sum form via Theorem 2.4. Thus, even though the number of alignments is exponential in the length of the acoustic and text input $\mathcal{O}(|x|+|y|)$, the likelihood and its gradient can be computed in time linear to the size of the lattice $\mathcal{O}(|x||y|)$. In practice, most implementations of CTC and RNN-T compute likelihoods in the log space for numerical stability, which corresponds to dynamic programming using a log semiring (cf. Definition 2.7). More broadly, Eqs. (5) and (6) can be recognized as specific instances of the more general Eq. (7).

$$P(y|x) = \sum_{\pi \in \mathcal{A}(x,y)} \prod_i P(\pi_i | x). \tag{7}$$

While we focus on CTC and RNN-T in our paper, other examples of neural speech recognition lattices that use dynamic programming to efficiently marginalize over an exponential number of alignments include Auto Segmentation Criterion (Collobert et al., 2016), Lattice-Free MMI (Povey et al., 2016), and the Recurrent Neural Aligner (Sak et al., 2017).

## 4  IMPLEMENTATION OF THE SEMIRING FRAMEWORK

While the mathematical details of the entropy semiring have been known for more than two decades, it is highly non-trivial to implement them in neural speech recognition models. This is in part because modern deep learning models operate on much longer sequences and have to compute gradients, and in part because of intricate implementation details like applying loop skewing on the RNN-T lattice (Bagby et al., 2018). One of the main contributions of our work is to make an open-source implementation of CTC and RNN-T in the semiring framework available to the research community (cf. Supplementary Material). Below, we introduce two variants of the entropy semiring that are designed for numerical stability and parallelism.

## 4.1 NUMERICAL STABILITY

A numerically stable implementation of the entropy semiring has to avoid the naive multiplication of two small numbers, and do it in log space instead. As such, the mathematical formulation provided in Definition 2.9 is not actually numerically stable because it involves multiplying $p_1$ and $p_2 \log p_2$ for two small numbers $p_1, p_2$. Notice that applying a log morphism on just the first argument of the dual number is not sufficient to ensure numerical stability in the backward pass because $\lim_{p \to 0} \frac{\partial}{\partial p} p \log p = \lim_{p \to 0} 1 + \log p = -\infty$.

We take care of numerical stability in both the forward and backward passes by applying a log morphism on both arguments of the dual number in the semiring. For the second argument, the log morphism has to be applied on the entropy $-p \log p$ since we cannot take the logarithm of negative numbers. This results in the following variant of the entropy semiring.

**Definition 4.1.** The *log entropy semiring* can be defined as such:

$$w(e) = \langle \log p(e), \log(-p(e) \log p(e)) \rangle,$$

$$\langle a, b \rangle \oplus \langle c, d \rangle = \langle \log(e^a + e^c), \log(e^b + e^d) \rangle,$$

$$\langle a, b \rangle \otimes \langle c, d \rangle = \langle a + c, \log(e^{a+d} + e^{b+c}) \rangle,$$

$$\bar{0} = \langle -\infty, -\infty \rangle, \bar{1} = \langle 0, -\infty \rangle.$$

## 4.2 ENTROPY SEMIRING FOR DISTILLATION

Minimizing the negative log likelihood of a parameterized model is equivalent to minimizing the KL divergence between the empirical data distribution and the modeled distribution. Knowledge distillation is a technique that proposes to use soft labels from the output of a teacher model in the place of hard labels in the ground truth data (Hinton et al., 2015). It is often implemented as a weighted sum of two different KL divergences: one between the empirical data distribution and the model (hard labels), and one between the teacher distribution and the model (soft labels).

$$\mathcal{L}_{\text{distill}} = KL\left(P_{\text{empirical}}||P_{\text{student}}\right) + \alpha_{\text{distill}} KL\left(P_{\text{teacher}}||P_{\text{student}}\right). \tag{8}$$

We can re-write the distillation loss in Eq. (8) as a sum of three different terms.

$$\mathcal{L}_{\text{distill}} = -\log P_{\text{student}} + \alpha_{\text{distill}} \left[P_{\text{teacher}} \log P_{\text{teacher}} - P_{\text{teacher}} \log P_{\text{student}}\right]. \tag{9}$$

Notice that the first term can be computed with the log semiring, while the second and third term can be computed with the log entropy semiring. But instead of doing three separate forward passes, we can compute the distillation loss using a single forward pass by concatenating them to form a new semiring weighted by four real number values $w \colon E \to \mathbb{R}^4$.

**Definition 4.2.** The *log reverse-KL semiring* can be defined as such:

$$w(e) = \langle \log p(e), \log q(e), \log(-q(e) \log q(e)), \log(-q(e) \log p(e)) \rangle,$$

$$\langle a, b, c, d \rangle \oplus \langle f, g, h, i \rangle = \langle \log(e^a + e^f), \log(e^b + e^g), \log(e^c + e^h), \log(e^d + e^i) \rangle,$$

$$\langle a, b, c, d \rangle \otimes \langle f, g, h, i \rangle = \langle a + f, b + g, \log(e^{b+h} + e^{c+g}), \log(e^{b+i} + e^{d+g}) \rangle,$$

$$\bar{0} = \langle -\infty, -\infty, -\infty, -\infty \rangle, \bar{1} = \langle 0, 0, -\infty, -\infty \rangle.$$

## 5 APPLICATIONS

Below, we study two applications of the entropy semiring for neural speech recognition.

## 5.1 ENTROPY REGULARIZATION

**Motivation** Because there is an exponential number of alignments, it is difficult in general for an optimization algorithm to settle on the optimal alignment given no explicit supervision. Instead, once a set of feasible alignments has been found during training, it tends to dominate, and error signals concentrate around the vicinity of such alignments. It has been experimentally observed that both CTC and RNN-T tend to produce highly peaky and over-confident distributions, and converge

Table 1: Word Error Rate with and without Entropy Regularization.

| Model | #Params | Method | Dev-Clean | Dev-Other | Test-Clean | Test-Other |
|---|---|---|---|---|---|---|
| CTC LSTM | 22M | Baseline | $7.7 \pm 0.3$ | $21.9 \pm 0.7$ | $7.7 \pm 0.3$ | $21.8 \pm 0.6$ |
| CTC LSTM | 22M | Ent | $\mathbf{7.3 \pm 0.3}$ | $\mathbf{20.7 \pm 0.7}$ | $\mathbf{7.2 \pm 0.3}$ | $\mathbf{20.8 \pm 0.6}$ |
| CTC Conformer | 9M | Baseline | $\mathbf{3.9 \pm 0.2}$ | $10.2 \pm 0.4$ | $\mathbf{4.1 \pm 0.2}$ | $10.2 \pm 0.4$ |
| CTC Conformer | 9M | Ent | $\mathbf{3.9 \pm 0.2}$ | $\mathbf{9.9 \pm 0.4}$ | $\mathbf{4.1 \pm 0.2}$ | $\mathbf{9.9 \pm 0.4}$ |
| RNN-T LSTM | 25M | Baseline | $7.8 \pm 0.4$ | $23.6 \pm 0.8$ | $7.4 \pm 0.3$ | $24.0 \pm 0.8$ |
| RNN-T LSTM | 25M | Ent | $\mathbf{7.4 \pm 0.3}$ | $\mathbf{22.5 \pm 0.8}$ | $\mathbf{7.2 \pm 0.3}$ | $\mathbf{23.1 \pm 0.7}$ |
| RNN-T Conformer | 10M | Baseline | $\mathbf{2.5 \pm 0.2}$ | $6.7 \pm 0.3$ | $2.8 \pm 0.2$ | $6.7 \pm 0.3$ |
| RNN-T Conformer | 10M | Ent | $\mathbf{2.5 \pm 0.2}$ | $\mathbf{6.5 \pm 0.3}$ | $\mathbf{2.7 \pm 0.2}$ | $\mathbf{6.5 \pm 0.3}$ |

towards local optima (Miao et al., 2015; Liu et al., 2018; Yu et al., 2021). There is also theoretical analysis that suggests that this phenomenon is to some extent inevitable, and a direct result of the training criterion (Zeyer et al., 2021; Blondel et al., 2021). We propose to ameliorate this problem via an entropy regularization mechanism that penalizes the negative entropy of the alignment distribution, which encourages the exploration of more alignment paths during training.

$$\mathcal{L}_{\text{Ent}} = \sum_{\pi \in \mathcal{A}(x,y)} - \log P_{\text{model}}(\pi) + \alpha_{\text{Ent}} P_{\text{model}}(\pi) \log P_{\text{model}}(\pi). \tag{10}$$

**Experimental Setup** We experimented with models using non-causal LSTM and Conformer (Gulati et al., 2020) encoders on the Librispeech dataset (Panayotov et al., 2015). The acoustic input is processed as 80-dimensional log-mel filter bank coefficients using short-time Fourier transform windows of size 25ms and stride 10ms. All models are trained with Adam using the optimization schedule specified in Vaswani et al. (2017), with a 10k warmup, batch size 2048, and a peak learning rate of 0.002. The LSTM encoders have 4 bi-directional layers with cell size 512 and are trained for 100k steps, while the Conformer encoders have 16 full-context attention layers with model dimension 144 and are trained for 400k steps. Decoding for all models is done with beam search, with the CTC decoders using a beam width of 16, and the RNN-T decoders using a beam width of 8 and a 1-layer LSTM with cell size 320. All models use a WordPiece tokenizer with a vocabulary size of 1024. $\alpha_{\text{Ent}}$ was selected via a grid search on $\{0.01, 0.001\}$. We report 95% confidence interval estimates for the WER following Vilar (2008).

**Results** We see from Table 1 that adding entropy regularization improves WER performance over the baseline model in almost all cases. On LSTMs, the improvements were the biggest, for example there was a big WER reduction of 6.5% in the Test-Clean case for the CTC LSTM from 7.7% to 7.2%. The improvements were smaller for the Conformers, for example with the WER remaining the same on Dev-Clean and dropping by an absolute 0.1% on Test-Clean for the RNN-T model. In general, we find that the improvements are the biggest when the baseline performance is the worst, indicating that entropy regularization is the most effective when the baseline model has converged sub-optimally and has under-utilized its model capacity. We provide visualizations and further qualitative analysis in Appendix D.

## 5.2 RNN-T Distillation

**Motivation** Distillation is a technique that uses a pre-trained teacher model to aid in the training of a student model. It can be especially helpful in the semi-supervised setting where a teacher model is trained on a small set of labeled data and then used to generate pseudo-labels for a larger set of unlabeled data (Park et al., 2020). A student model is then trained using the standard NLL loss on the combined set of labeled and pseudo-labeled data. We refer to this process as *hard* distillation.

$$\mathcal{L}_{\text{hard}} = - \log P_{\text{student}}(\hat{y} = l_{\text{teacher}}|x). \tag{11}$$

In addition to the hard pseudo-labels, we can use the soft logits from the teacher model to provide a better training signal to the student model. Prior work for RNN-T models has implemented this by

summing up the KL divergence between the state-wise posteriors of both models (Kurata & Saon, 2020; Panchapagesan et al., 2021). We refer to this process as *soft* distillation.

$$\mathcal{L}_{\text{soft}} = \mathcal{L}_{\text{hard}} + \alpha_{\text{state}} KL_{\text{state}}.$$

$$KL_{\text{state}} = \sum_{t,u,v} P_{\text{teacher}}(\hat{y}_{t+u} = l_v | x_{1:t}, y_{1:u-1}) \log \frac{P_{\text{teacher}}(\hat{y}_{t+u} = l_v | x_{1:t}, y_{1:u-1})}{P_{\text{student}}(\hat{y}_{t+u} = l_v | x_{1:t}, y_{1:u-1})}. \quad (12)$$

Streaming models have to carefully balance the competing objectives of reducing emission latency and improving WER accuracy. Alignments that delay emission can leverage more information to perform its prediction, but at the cost of latency. Because both the hard and soft distillation objectives operate in token space, they neglect potentially helpful alignment information from the teacher model. Prior work has found that alignment information from a good pre-trained model can help guide attention-based encoder-decoder models to achieve both good latency and accuracy (Inaguma & Kawahara, 2021). We propose to add a KL divergence loss for the sequence-wise posteriors of both models to incorporate alignment information into the distillation objective. Because this technique is based on the use of a semiring (cf. Section 4.2), we refer to it as *semiring* distillation.

$$\mathcal{L}_{\text{semiring}} = \mathcal{L}_{\text{soft}} + \alpha_{\text{seq}} KL_{\text{seq}}.$$

$$KL_{\text{seq}} = \sum_{\pi \in \mathcal{A}_{\text{RNN-T}}(x,y)} P_{\text{teacher}}(\pi) \log \frac{P_{\text{teacher}}(\pi)}{P_{\text{student}}(\pi)}. \quad (13)$$

**Experimental Setup** We set up a semi-supervised learning scenario by using Libri-Light as the unlabeled dataset (Kahn et al., 2020) and Librispeech as the labeled dataset. First, we prepare pseudo-labels for Libri-Light from Zhang et al. (2020b), where an iterative hard distillation process is performed with a non-causal 1.0B Conformer model to arrive at high quality pseudo-labels. Then, we train a teacher model by randomly sampling batches from Libri-Light and Librispeech in a 90:10 mix. This teacher model is used as our hard distillation baseline, and also used to train a student model via soft and semiring distillation for comparison. Both teacher and student models use the same 0.6B causal Conformer architecture as in Chiu et al. (2022) with self-supervised pre-training on Libri-Light via a random projection quantizer. Like Hwang et al. (2022), the student model has FreqAug applied to it for data augmentation. Both models were trained for 160k steps with Adam using the same optimization schedule as in Vaswani et al. (2017), but with a peak learning rate of 0.0015, a 5k warmup, and batch size 256. The RNN-T decoder uses a 2-layer LSTM with cell size 1200, beam width 8, and a WordPiece tokenizer with a vocabulary size of 1024. We do a grid search for both $\alpha_{\text{state}}$ and $\alpha_{\text{seq}}$ over $\{0.01, 0.001\}$. We report 95% confidence interval estimates for the WER following Vilar (2008).

**Results** We report our results in Table 2. We see that soft distillation results in substantial WER improvements over the hard distillation baseline, reducing WER on the Test-Clean set by 22% from 2.7% to 2.1% and on the Test-Other set by 17% from 6.4% to 5.3%. Semiring distillation results in small WER improvements over soft distillation, further reducing WER by an absolute 0.1% on all the Dev and Test sets. We further do an ablation study with semiring distillation using $\alpha_{\text{state}} = 0.0$, and find that alignment distillation alone, while under-performing soft distillation, still results in significant WER improvements over the hard distillation baseline. These results suggest that while a good measure of uncertainty in the token space helps improve accuracy more than a corresponding measure of uncertainty in the alignment space, these two measures of uncertainty are complementary to each other and should be used jointly for best results.

We do relative latency measurements between two models following Chiu et al. (2022). Start and end times of every word in the output hypotheses are first calculated, and then used to compute the average word timing difference between matching words from the hypotheses of both models. Interestingly, we see that even though the soft distillation model obtained better WER accuracy, this came at a slight cost to emission latency (+2.7ms) relative to the baseline hard distillation model. On the other hand, the $\alpha_{\text{state}} = 0.0$ semiring distillation ablation performed significantly better in terms of relative latency ($-65.7$ms), indicating that alignment information from the teacher helped the student learn to emit tokens faster. By combining both state-wise and sequence-wise posteriors, the semiring distillation model improved on both emission latency and WER accuracy compared to the hard or soft distillation model.

Table 2: Self-Distillation for a 0.6B Causal Conformer RNN-T Model.

| Distillation Method | Relative Latency (ms) | Dev-Clean | Dev-Other | Test-Clean | Test-Other |
|---|---|---|---|---|---|
| Hard Distillation | 0.0 | $2.5 \pm 0.2$ | $6.8 \pm 0.4$ | $2.7 \pm 0.2$ | $6.4 \pm 0.3$ |
| Soft Distillation | $+2.7$ | $1.9 \pm 0.1$ | $5.3 \pm 0.3$ | $2.1 \pm 0.2$ | $5.3 \pm 0.3$ |
| Semiring Distillation | | | | | |
| with $\alpha_{state} = 0.0$ | $\mathbf{-65.7}$ | $2.0 \pm 0.2$ | $5.5 \pm 0.3$ | $2.1 \pm 0.2$ | $5.8 \pm 0.3$ |
| with $\alpha_{state} = 0.001$ | $-5.2$ | $\mathbf{1.8 \pm 0.1}$ | $\mathbf{5.2 \pm 0.3}$ | $\mathbf{2.0 \pm 0.2}$ | $\mathbf{5.2 \pm 0.3}$ |

Table 3: Comparison with Prior Work on Librispeech in the Streaming Setting. Libri-Light: uses unlabeled data from Libri-Light. Lookahead: the model uses a limited amount of future context.

| Prior Work | Libri-Light | Lookahead | #Params | Dev-Clean | Dev-Other | Test-Clean | Test-Other |
|---|---|---|---|---|---|---|---|
| Zhang et al. (2020a) | No | No | - | - | - | 4.2 | 11.3 |
| Zhang et al. (2020a) | No | Yes | - | - | - | 3.6 | 10.0 |
| Yu et al. (2020) | No | No | 30M | - | - | 3.7 | 9.2 |
| Cao et al. (2021) | No | No | - | 3.2 | 8.5 | 3.5 | 8.7 |
| Hwang et al. (2022) | Yes | No | 0.2B | 4.0 | 9.4 | 4.4 | 8.6 |
| Moritz et al. (2020) | No | No | - | 2.9 | 8.1 | 3.2 | 8.0 |
| Yu et al. (2021) | No | No | - | - | - | 3.1 | 7.5 |
| Moritz et al. (2020) | No | Yes | - | 2.7 | 7.1 | 2.8 | 7.2 |
| Chiu et al. (2022) | Yes | No | 0.6B | 2.5 | 6.9 | 2.8 | 6.6 |
| Shi et al. (2021) | No | Yes | 80M | - | - | 2.4 | 6.1 |
| Our work | Yes | No | 0.6B | **1.8** | **5.2** | **2.0** | **5.2** |

Finally, we compare the performance of our semiring-distilled model with prior work for Librispeech in the streaming setting in Table 3. To the best of our knowledge, our work has achieved a new state-of-the-art on this benchmark, without using any future context in making its predictions.

## 6 CONCLUSION

Across multiple machine learning applications in speech and natural language processing, computational biology, and even computer vision, there has been a paradigm shift towards sequence-to-sequence modeling using Transformer-based architectures. Most learning objectives on these models do not take advantage of the structured nature of the inputs and outputs. This is in stark contrast to the rich literature of sequential modeling approaches based on weighted finite-state transducers from before the deep learning renaissance (Eisner, 2002; Mohri et al., 2002; Cortes et al., 2004).

Our work draws upon these pre deep learning era approaches to introduce alignment-based supervision for neural ASR in the form of regularization and distillation. We believe that applying similar semiring based techniques to supervise the alignment between data of different modalities or domains will lead to better learning representations. For example, recent work has found that image-text models do not have good visio-linguistic compositional reasoning abilities (Thrush et al., 2022). This problem can potentially be addressed by learning a good alignment model between a natural language description of an object and its corresponding pixel-level image representation. Another area of future work can be learning to synchronize audio and visual input streams to solve tasks with multiple output streams like overlapping speech recognition or diarization.

Finally, we are excited about the gamut of new ASR research that will arise from plugging different semirings into our CTC and RNN-T semiring framework implementation. For example, even though most mainstream approaches use the log semiring for training and decoding, a differentiable relaxation akin to Cuturi & Blondel (2017) might yield surprising findings. Another promising direction for future work can involve calculating the mean or variance of hypothesis lengths for diagnostic or regularization purposes.

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

# APPENDIX

The Appendix is organized as follows: Appendix A discusses other related work in the literature. Appendix B provides proofs that the semirings defined in our paper are actual semirings. Appendix C provides extended derivations for Examples 2.8 and 2.10. Appendix D showcases visualizations and further qualitative analysis on the effects of entropy regularization.

## A    RELATED WORK

Cortes et al. (2004) showed how weighted transducers can be used to form rational kernels for support vector machines, thereby expanding their use to classification problems with variable-length inputs. Thereafter, Nelken & Shieber (2006) demonstrated a rational kernel with the entropy semiring. The use of semirings has been well-established in the conditional random fields literature: replacing the max-plus algebra with the sum-product algebra is a way to derive the forward-backward algorithm from Viterbi (Bishop & Nasrabadi, 2006; Wainwright et al., 2008). Li & Eisner (2009) studied first-order and second-order expectation semirings with applications to machine translation. Eisner (2016) discussed how the inside-outside algorithm can be derived simply from backpropagation and automatic differentiation. Liu et al. (2018) proposed a different CTC-specific entropy regularization objective that is based on a similar motivation of avoiding bad alignments. Cuturi & Blondel (2017); Mensch & Blondel (2018); Blondel et al. (2021) explore various differentiable relaxations for dynamic programming based losses in deep learning. Hannun et al. (2020) proposed differentiable weighted finite state transducers, which focused on plug-and-play graphs that support the log and tropical semirings, as opposed to our work, which focuses on plug-and-play semirings that support the CTC and RNN-T graphs. Because our graphs are fixed and coded in Tensorflow itself, we are able to perform graph optimizations (for example, loop-skewing on the RNN-T lattice), compiler optimizations (for example, the XLA compiler for Tensorflow produces production-quality binaries for a variety of software and hardware architectures), and even semiring optimizations (for example, our proposed variants of the entropy semiring that are more numerically stable and highly parallel), thereby ensuring an especially efficient implementation for neural speech recognition. The use of both state-wise and sequence-wise posteriors in semiring distillation is similar to the direct sum entropy introduced in Farinhas et al. (2021), which is a mix of both discrete entropy and differential entropy.

# B    PROOF OF SEMIRING

Below, we prove that the probability, log, entropy, log entropy, log reverse-KL semirings defined in our paper are actual semirings according to Definition 2.2. Each proof is provided in four parts:

1. $\otimes$ forms a commutative monoid with identity $\bar{0}$.

2. $\oplus$ forms a monoid with identity $\bar{1}$.

3. $\otimes$ distributes over $\oplus$.

4. $\bar{0}$ annihilates $\otimes$.

## B.1    PROOF THAT THE PROBABILITY SEMIRING IS A SEMIRING

**Definition 2.5.** The *probability semiring* can be defined as such: $w(e) = p(e)$ where $p\colon E \to [0, 1]$ is the probability function, $\oplus = +, \otimes = \times, \bar{0} = 0, \bar{1} = 1$.

**Proof.**

1. $a \oplus b$
   $= a + b$
   $= b + a$
   $= b \oplus a.$

   $a \oplus (b \oplus c)$
   $= a + (b + c)$
   $= (a + b) + c$
   $= (a \oplus b) \oplus c.$

   $a \oplus \bar{0}$
   $= a + 0$
   $= a.$

2. $a \otimes (b \otimes c)$
   $= a \times (b \times c)$
   $= (a \times b) \times c$
   $= (a \otimes b) \otimes c.$

   $a \otimes \bar{1}$
   $= a \times 1$
   $= a.$

   $\bar{1} \otimes a$
   $= 1 \times a$
   $= a.$

3. $a \otimes (b \oplus c)$
   $= a \times (b + c)$
   $= (a \times b) + (a \times c)$
   $= (a \otimes b) \oplus (a \otimes c).$

   $(b \oplus c) \otimes a$
   $= (b + c) \times a$
   $= (b \times a) + (c \times a)$
   $= (b \otimes a) \oplus (c \otimes a).$

4. $a \otimes \bar{0}$
   $= a \times 0$
   $= 0$
   $= \bar{0}.$

   $\bar{0} \otimes a$

$$= 0 \times a$$
$$= 0$$
$$= \bar{0}.$$

## B.2 Proof that the Log Semiring is a Semiring

**Definition 2.7.** The *log semiring* can be defined as such: $w(e) = \log p(e)$, $a \oplus b = \log(e^a + e^b)$, $a \otimes b = a + b$, $\bar{0} = -\infty$, $\bar{1} = 0$.

**Proof.**

1. $a \oplus b$
   $= \log(e^a + e^b)$
   $= \log(e^b + e^a)$
   $= b \oplus a.$

   $a \oplus (b \oplus c)$
   $= a \oplus \log(e^b + e^c)$
   $= \log(e^a + e^b + e^c)$
   $= \log(e^a + e^b) \oplus c$
   $= (a \oplus b) \oplus c.$

   $a \oplus \bar{0}$
   $= \log(e^a + e^{-\infty})$
   $= a.$

2. $a \otimes (b \otimes c)$
   $= a + (b + c)$
   $= (a + b) + c$
   $= (a \otimes b) \otimes c.$

   $a \otimes \bar{1}$
   $= a + 0$
   $= a.$

   $\bar{1} \otimes a$
   $= 0 + a$
   $= a.$

3. $a \otimes (b \oplus c)$
   $= a + \log(e^b + e^c)$
   $= \log(e^{a+b} + e^{a+c})$
   $= (a \otimes b) \oplus (a \otimes c).$

   $(b \oplus c) \otimes a$
   $= \log(e^b + e^c) + a$
   $= \log(e^{b+a} + e^{c+a})$
   $= (b \otimes a) \oplus (c \otimes a).$

4. $a \otimes \bar{0}$
   $= a - \infty$
   $= -\infty$
   $= \bar{0}.$

   $\bar{0} \otimes a$
   $= -\infty + a$
   $= -\infty$
   $= \bar{0}.$

### B.3 PROOF THAT THE ENTROPY SEMIRING IS A SEMIRING

**Definition 2.9.** The *entropy semiring* can be defined as such:

$$w(e) = \langle p(e), p(e) \log p(e) \rangle,$$

$$\langle a, b \rangle \oplus \langle c, d \rangle = \langle a + c, b + d \rangle,$$

$$\langle a, b \rangle \otimes \langle c, d \rangle = \langle ac, ad + bc \rangle,$$

$$\bar{0} = \langle 0, 0 \rangle, \bar{1} = \langle 1, 0 \rangle.$$

**Proof.**

1. $\langle a, b \rangle \oplus \langle c, d \rangle$
   $= \langle a + c, b + d \rangle$
   $= \langle c + a, d + b \rangle$
   $= \langle c, d \rangle \oplus \langle a, b \rangle.$

   $\langle a, b \rangle \oplus (\langle c, d \rangle \oplus \langle f, g \rangle)$
   $= \langle a, b \rangle \oplus \langle c + f, d + g \rangle$
   $= \langle a + c + f, b + d + g \rangle$
   $= \langle a + c, b + d \rangle \oplus \langle f, g \rangle$
   $= (\langle a, b \rangle \oplus \langle c, d \rangle) \oplus \langle f, g \rangle.$

   $\langle a, b \rangle \oplus \bar{0}$
   $= \langle a + 0, b + 0 \rangle$
   $= \langle a, b \rangle.$

2. $\langle a, b \rangle \otimes (\langle c, d \rangle \otimes \langle f, g \rangle)$
   $= \langle a, b \rangle \otimes \langle cf, cg + df \rangle$
   $= \langle acf, acg + adf + bcf \rangle$
   $= \langle ac, ad + bc \rangle \otimes \langle f, g \rangle$
   $= (\langle a, b \rangle \otimes \langle c, d \rangle) \otimes \langle f, g \rangle.$

   $\langle a, b \rangle \otimes \bar{1}$
   $= \langle a \times 1, a \times 0 + b \times 1 \rangle$
   $= \langle a, b \rangle.$

   $\bar{1} \otimes \langle a, b \rangle$
   $= \langle 1 \times a, 0 \times a + 1 \times b \rangle$
   $= \langle a, b \rangle.$

3. $\langle a, b \rangle \otimes (\langle c, d \rangle \oplus \langle f, g \rangle)$
   $= \langle a, b \rangle \otimes \langle c + f, d + g \rangle$
   $= \langle ac + af, ad + ag + bc + bf \rangle$
   $= \langle ac, ad + bc \rangle \oplus \langle af, ag + bf \rangle$
   $= (\langle a, b \rangle \otimes \langle c, d \rangle) \oplus (\langle a, b \rangle \otimes \langle f, g \rangle).$

   $(\langle c, d \rangle \oplus \langle f, g \rangle) \otimes \langle a, b \rangle$
   $= \langle c + f, d + g \rangle \otimes \langle a, b \rangle$
   $= \langle ca + fa, cb + fb + da + ga \rangle$
   $= \langle ca, cb + da \rangle \oplus \langle fa, fb + ga \rangle$
   $= (\langle c, d \rangle \otimes \langle a, b \rangle) \oplus (\langle f, g \rangle) \otimes \langle a, b \rangle).$

4. $\langle a, b \rangle \otimes \bar{0}$
   $= \langle a \times 0, a \times 0 + b \times 0 \rangle$
   $= \langle 0, 0 \rangle$
   $= \bar{0}.$

   $\bar{0} \otimes \langle a, b \rangle$
   $= \langle 0 \times a, 0 \times a + 0 \times b \rangle$

$$= \langle 0, 0 \rangle$$
$$= \bar{0}.$$

## B.4   Proof that the Log Entropy Semiring is a Semiring

**Definition 4.1.** The *log entropy semiring* can be defined as such:

$$w(e) = \langle \log p(e), \log(-p(e) \log p(e)) \rangle,$$

$$\langle a, b \rangle \oplus \langle c, d \rangle = \langle \log(e^a + e^c), \log(e^b + e^d) \rangle,$$

$$\langle a, b \rangle \otimes \langle c, d \rangle = \langle a + c, \log(e^{a+d} + e^{b+c}) \rangle,$$

$$\bar{0} = \langle -\infty, -\infty \rangle, \bar{1} = \langle 0, -\infty \rangle.$$

**Proof.**

1. $\langle a, b \rangle \oplus \langle c, d \rangle$
   $= \langle \log(e^a + e^c), \log(e^b + e^d) \rangle$
   $= \langle \log(e^c + e^a), \log(e^d + e^b) \rangle$
   $= \langle c, d \rangle \oplus \langle a, b \rangle.$

   $\langle a, b \rangle \oplus (\langle c, d \rangle \oplus \langle f, g \rangle)$
   $= \langle a, b \rangle \oplus \langle \log(e^c + e^f), \log(e^d + e^g) \rangle$
   $= \langle \log(e^a + e^c + e^f), \log(e^b + e^d + e^g) \rangle$
   $= \langle \log(e^a + e^c), \log(e^b + e^d) \rangle \oplus \langle f, g \rangle$
   $= (\langle a, b \rangle \oplus \langle c, d \rangle) \oplus \langle f, g \rangle.$

   $\langle a, b \rangle \oplus \bar{0}$
   $= \langle \log(e^a + e^{-\infty}), \log(e^b + e^{-\infty}) \rangle$
   $= \langle a, b \rangle.$

2. $\langle a, b \rangle \otimes (\langle c, d \rangle \otimes \langle f, g \rangle)$
   $= \langle a, b \rangle \otimes \langle c + f, \log(e^{c+g} + e^{d+f}) \rangle$
   $= \langle a + c + f, \log(e^{a+c+g} + e^{a+d+f} + e^{b+c+f}) \rangle$
   $= \langle a + c, \log(e^{a+d} + e^{b+c}) \rangle \otimes \langle f, g \rangle$
   $= (\langle a, b \rangle \otimes \langle c, d \rangle) \otimes \langle f, g \rangle.$

   $\langle a, b \rangle \otimes \bar{1}$
   $= \langle a + 0, \log(e^{a-\infty} + e^{b+0}) \rangle$
   $= \langle a, b \rangle.$

   $\bar{1} \otimes \langle a, b \rangle$
   $= \langle 0 + a, \log(e^{-\infty+a} + e^{0+b}) \rangle$
   $= \langle a, b \rangle.$

3. $\langle a, b \rangle \otimes (\langle c, d \rangle \oplus \langle f, g \rangle)$
   $= \langle a, b \rangle \otimes \langle \log(e^c + e^f), \log(e^d + e^g) \rangle$
   $= \langle a + \log(e^c + e^f), \log(e^{a+d} + e^{a+g} + e^{b+c} + e^{b+f}) \rangle$
   $= \langle a + c, \log(e^{a+d} + e^{b+c}) \rangle \oplus \langle a + f, \log(e^{a+g} + e^{b+f}) \rangle$
   $= (\langle a, b \rangle \otimes \langle c, d \rangle) \oplus (\langle a, b \rangle \otimes \langle f, g \rangle).$

   $(\langle c, d \rangle \oplus \langle f, g \rangle) \otimes \langle a, b \rangle$
   $= \langle \log(e^c + e^f), \log(e^d + e^g) \rangle \otimes \langle a, b \rangle$
   $= \langle \log(e^c + e^f) + a, \log(e^{c+b} + e^{f+b} + e^{d+a} + e^{g+a}) \rangle$
   $= \langle c + a, \log(e^{c+b} + e^{d+a}) \rangle \oplus \langle f + a, \log(e^{f+b} + e^{g+a}) \rangle$
   $= (\langle c, d \rangle \otimes \langle a, b \rangle) \oplus (\langle f, g \rangle \otimes \langle a, b \rangle).$

4. $\langle a, b \rangle \otimes \bar{0}$
   $= \langle a - \infty, \log(e^{a-\infty} + e^{b-\infty}) \rangle$
   $= \langle -\infty, -\infty \rangle$
   $= \bar{0}.$

$\bar{0} \otimes \langle a, b \rangle$
$= \langle -\infty + a, \log(e^{-\infty+a} + e^{-\infty+b}) \rangle$
$= \langle -\infty, -\infty \rangle$
$= \bar{0}.$

## B.5 Proof that the Log Reverse-KL Semiring is a Semiring

**Definition 4.2.** The *log reverse-KL semiring* can be defined as such:

$w(e) = \langle \log p(e), \log q(e), \log(-q(e) \log q(e)), \log(-q(e) \log p(e)) \rangle,$

$\langle a, b, c, d \rangle \oplus \langle f, g, h, i \rangle = \langle \log(e^a + e^f), \log(e^b + e^g), \log(e^c + e^h), \log(e^d + e^i) \rangle,$

$\langle a, b, c, d \rangle \otimes \langle f, g, h, i \rangle = \langle a + f, b + g, \log(e^{b+h} + e^{c+g}), \log(e^{b+i} + e^{d+g}) \rangle,$

$\bar{0} = \langle -\infty, -\infty, -\infty, -\infty \rangle, \bar{1} = \langle 0, 0, -\infty, -\infty \rangle.$

**Proof.**

1. $\langle a, b, c, d \rangle \oplus \langle f, g, h, i \rangle$
   $= \langle \log(e^a + e^f), \log(e^b + e^g), \log(e^c + e^h), \log(e^d + e^i) \rangle$
   $= \langle \log(e^f + e^a), \log(e^g + e^b), \log(e^h + e^c), \log(e^i + e^d) \rangle$
   $= \langle f, g, h, i \rangle \oplus \langle a, b, c, d \rangle.$

   $\langle a, b, c, d \rangle \oplus (\langle f, g, h, i \rangle \oplus \langle j, k, l, m \rangle)$
   $= \langle a, b, c, d \rangle \oplus \langle \log(e^f + e^j), \log(e^g + e^k), \log(e^h + e^l), \log(e^i + e^m) \rangle$
   $= \langle \log(e^a + e^f + e^j), \log(e^b + e^g + e^k), \log(e^c + e^h + e^l), \log(e^d + e^i + e^m) \rangle$
   $= \langle \log(e^a + e^f), \log(e^b + e^g), \log(e^c + e^h), \log(e^d + e^i) \rangle \oplus \langle j, k, l, m \rangle$
   $= (\langle a, b, c, d \rangle \oplus \langle f, g, h, i \rangle) \oplus \langle j, k, l, m \rangle.$

   $\langle a, b, c, d \rangle \oplus \bar{0}$
   $= \langle \log(e^a + e^{-\infty}), \log(e^b + e^{-\infty}), \log(e^c + e^{-\infty}), \log(e^d + e^{-\infty}) \rangle$
   $= \langle a, b, c, d \rangle.$

2. $\langle a, b, c, d \rangle \otimes (\langle f, g, h, i \rangle \otimes \langle j, k, l, m \rangle)$
   $= \langle a, b, c, d \rangle \otimes \langle f + j, g + k, \log(e^{g+l} + e^{h+k}), \log(e^{g+m} + e^{i+k}) \rangle$
   $= \langle a + f + j, b + g + k, \log(e^{b+g+l} + e^{b+h+k} + e^{c+g+k}), \log(e^{b+g+m} + e^{b+i+k} + e^{d+g+k}) \rangle$
   $= \langle a + f, b + g, \log(e^{b+h} + e^{c+g}), \log(e^{b+i} + e^{d+g}) \rangle \otimes \langle j, k, l, m \rangle$
   $= (\langle a, b, c, d \rangle \otimes \langle f, g, h, i \rangle) \otimes \langle j, k, l, m \rangle.$

   $\langle a, b, c, d \rangle \otimes \bar{1}$
   $= \langle a + 0, b + 0, \log(e^{b-\infty} + e^{c+0}), \log(e^{b-\infty} + e^{d+0}) \rangle$
   $= \langle a, b, c, d \rangle.$

   $\bar{1} \otimes \langle a, b, c, d \rangle$
   $= \langle 0 + a, 0 + b, \log(e^{-\infty+b} + e^{0+c}), \log(e^{-\infty+b} + e^{0+d}) \rangle$
   $= \langle a, b, c, d \rangle.$

3. $\langle a, b, c, d \rangle \otimes (\langle f, g, h, i \rangle \oplus \langle j, k, l, m \rangle)$
   $= \langle a, b, c, d \rangle \otimes \langle \log(e^f + e^j), \log(e^g + e^k), \log(e^h + e^l), \log(e^i + e^m) \rangle$
   $= \langle a + \log(e^f + e^j), b + \log(e^g + e^k), \log(e^{b+h} + e^{b+l} + e^{c+g} + e^{c+k}), \log(e^{b+i} + e^{b+m} + e^{d+g} + e^{d+k}) \rangle$
   $= \langle a + f, b + g, \log(e^{b+h} + e^{c+g}), \log(e^{b+i} + e^{d+g}) \rangle \oplus \langle a + j, b + k, \log(e^{b+l} + e^{c+k}), \log(e^{b+m} + e^{d+k}) \rangle$
   $= (\langle a, b, c, d \rangle \otimes \langle f, g, h, i \rangle) \oplus (\langle a, b, c, d \rangle \otimes \langle j, k, l, m \rangle).$

   $(\langle f, g, h, i \rangle \oplus \langle j, k, l, m \rangle) \otimes \langle a, b, c, d \rangle$
   $= \langle \log(e^f + e^j), \log(e^g + e^k), \log(e^h + e^l), \log(e^i + e^m) \rangle \otimes \langle a, b, c, d \rangle$
   $= \langle \log(e^f + e^j) + a, \log(e^g + e^k) + b, \log(e^{g+c} + e^{k+c} + e^{h+b} + e^{l+b}), \log(e^{g+d} + e^{k+d} + e^{i+b} + e^{m+b}) \rangle$
   $= \langle f + a, g + b, \log(e^{g+c} + e^{h+b}), \log(e^{g+d} + e^{i+b}) \rangle \oplus \langle j + a, k + b, \log(e^{k+c} +$

$$e^{l+b}), \log(e^{k+d} + e^{m+b})\rangle$$
$$= (\langle f, g, h, i\rangle \otimes \langle a, b, c, d\rangle) \oplus (\langle j, k, l, m\rangle) \otimes \langle a, b, c, d\rangle).$$

4. $\langle a, b, c, d\rangle \otimes \bar{0}$
$$= \langle a - \infty, b - \infty, \log(e^{b-\infty} + e^{c-\infty}), \log(e^{b-\infty} + e^{d-\infty})\rangle$$
$$= \langle -\infty, -\infty, -\infty, -\infty\rangle$$
$$= \bar{0}.$$

$\bar{0} \otimes \langle a, b, c, d\rangle$
$$= \langle -\infty + a, -\infty + b, \log(e^{-\infty+b} + e^{-\infty+c}), \log(e^{-\infty+b} + e^{-\infty+d})\rangle$$
$$= \langle -\infty, -\infty, -\infty, -\infty\rangle$$
$$= \bar{0}.$$

## C  EXTENDED EXAMPLES

$$v_1 \xrightarrow{e_1} \quad \xrightarrow{e_3} v_4$$
$$v_3$$
$$v_2 \xrightarrow{e_2} \quad \xrightarrow{e_4} v_5$$

Figure 2: DAG in Example 2.6.

### C.1  DERIVATION FOR EXAMPLE 2.8

$$
\begin{aligned}
w(v_1) &= w(v_2) \\
&= 0. \\
w(v_3) &= (w(v_1) \otimes w(e_1)) \oplus (w(v_2) \otimes w(e_2)) \\
&= (0 + \log p(e_1)) \oplus (0 + \log p(e_2)) \\
&= \log p(e_1) \oplus \log p(e_2) \\
&= \log [p(e_1) + p(e_2)] . \\
w(v_4) &= w(v_3) \otimes w(e_3) \\
&= \log [p(e_1) + p(e_2)] \otimes \log p(e_3) \\
&= \log [p(e_1)p(e_3) + p(e_2)p(e_3)] . \\
w(v_5) &= w(v_3) \otimes w(e_4) \\
&= \log [p(e_1) + p(e_2)] \otimes \log p(e_4) \\
&= \log [p(e_1)p(e_4) + p(e_2)p(e_4)] . \\
C_{\log p(\pi)} &= w(v_4) \oplus w(v_5) \\
&= \log [p(e_1)p(e_3) + p(e_2)p(e_3)] \oplus \log [p(e_1)p(e_4) + p(e_2)p(e_4)] . \\
&= \log [p(e_1)p(e_3) + p(e_2)p(e_3) + p(e_1)p(e_4) + p(e_2)p(e_4)] .
\end{aligned}
$$

### C.2  DERIVATION FOR EXAMPLE 2.10

$$
\begin{aligned}
w(v_1) &= w(v_2) \\
&= \langle 1, 0 \rangle. \\
w(v_3) &= (w(v_1) \otimes w(e_1)) \oplus (w(v_2) \otimes w(e_2)) \\
&= (\langle 1, 0 \rangle \otimes \langle p(e_1), p(e_1) \log p(e_1) \rangle) \oplus (\langle 1, 0 \rangle + \langle p(e_2), p(e_2) \log p(e_2) \rangle) \\
&= \langle p(e_1), p(e_1) \log p(e_1) \rangle \oplus \langle p(e_2), p(e_2) \log p(e_2) \rangle \\
&= \langle p(e_1) + p(e_2), p(e_1) \log p(e_1) + p(e_2) \log p(e_2) \rangle. \\
w(v_4) &= w(v_3) \otimes w(e_3) \\
&= \langle p(e_1) + p(e_2), p(e_1) \log p(e_1) + p(e_2) \log p(e_2) \rangle \otimes \langle p(e_3), p(e_3) \log p(e_3) \rangle \\
&= \langle p(e_1)p(e_3) + p(e_2)p(e_3), p(e_1)p(e_3) \log p(e_1)p(e_3) + p(e_2)p(e_3) \log p(e_2)p(e_3) \rangle. \\
w(v_5) &= w(v_3) \otimes w(e_4) \\
&= \langle p(e_1) + p(e_2), p(e_1) \log p(e_1) + p(e_2) \log p(e_2) \rangle \otimes \langle p(e_4), p(e_4) \log p(e_4) \rangle \\
&= \langle p(e_1)p(e_4) + p(e_2)p(e_4), p(e_1)p(e_4) \log p(e_1)p(e_4) + p(e_2)p(e_4) \log p(e_2)p(e_4) \rangle. \\
C_{\langle p(\pi), p(\pi) \log p(\pi) \rangle} &= w(v_4) \oplus w(v_5) \\
&= \big\langle C_{p(\pi)}, \ p(e_1)p(e_3) \log [p(e_1)p(e_3)] + p(e_2)p(e_3) \log [p(e_2)p(e_3)] \\
&\quad + p(e_1)p(e_4) \log [p(e_1)p(e_4)] + p(e_2)p(e_4) \log [p(e_2)p(e_4)] \big\rangle.
\end{aligned}
$$

# D    VISUALIZATION FOR ENTROPY REGULARIZATION

A short example is picked at random from the Librispeech Test-Other set, which corresponds to the speech utterance *"That's what I believe young woman."* This example has an acoustic sequence length of 275 and a text sequence length of 8, which corresponds to a CTC lattice of size 275 by 17. Its waveform and log-mel spectrogram are respectively visualized in Fig. 3 and Fig. 4.

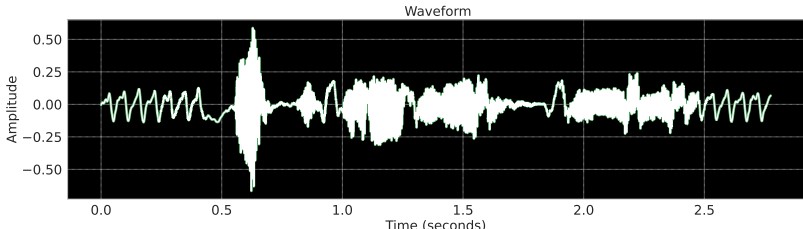

Figure 3: Visualization of waveform. The y-axis represents the amplitude of the signal, while the x-axis represents the time.

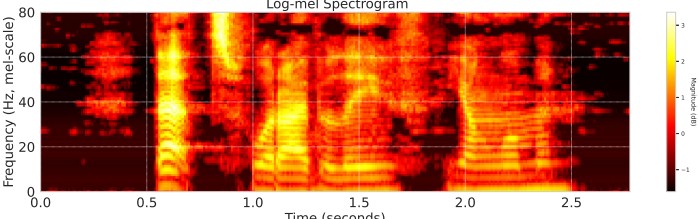

Figure 4: Visualization of log-mel spectrogram. The y-axis represents the frequency, while the x-axis represents the time. The color gradients represent the magnitude (in decibels) of the signal.

Fig. 5 visualizes the CTC lattices for this example using the LSTM models trained in Section 5.1 to showcase the effects of entropy regularization. Interestingly, the lattice on the right, which corresponds to higher alignment entropy, generally has higher "peaks" than the left lattice for both the blank labels and the acoustic regions corresponding to the non-blank labels. But these "peaks" are also wider, leading to an overall flatter and less peaky distribution. The visualization suggests that the surprising result of entropy regularization, which encourages the exploration of more alignment paths, is to concentrate, rather than diffuse, the probability mass around the right states in the lattice. In other words, instead of avoiding the local optima around the peaks, the model actually doubles down on them and widens the local exploration in their vicinity.

While this is helpful on average, it can also backfire. In Fig. 6, we visualize a pair of CTC lattices similar to Fig. 5, but for an utterance where the entropy regularized model has a higher WER compared to the NLL model. For this example, we see that the entropy regularized model has higher and wider probability masses at some erroneous states like $(\text{Audio} = 55, \text{Labels} = 7)$, i.e. it is actually more over-confident compared to the NLL model. This suggests that even though alignment entropy regularization is helpful on average, it might not be sufficient for good model calibration since more local exploration around existing bad optima can actually worsen the problem. Future work should investigate regularization mechanisms that encourage more global exploration further away from existing local optima in the alignment space.

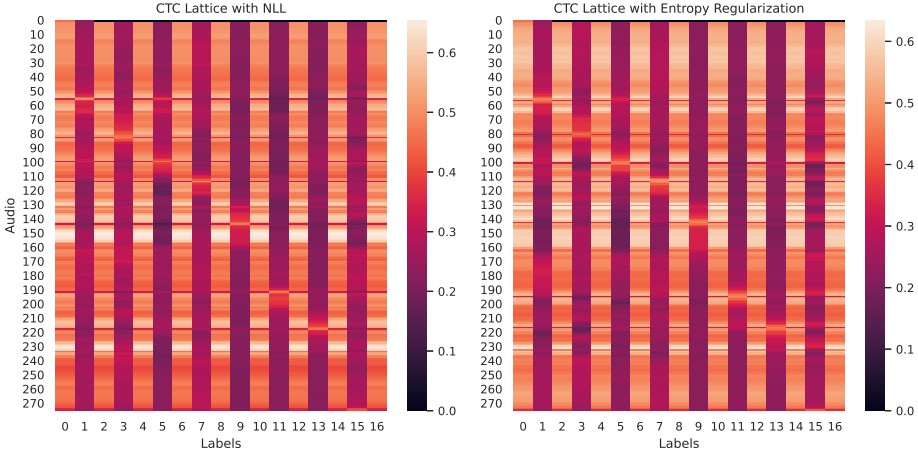

Figure 5: Visualization of CTC lattices. The left graph represents the CTC lattice trained with NLL, while the right graph represents the CTC lattice trained with entropy regularization. For both graphs, the y-axis represents the acoustic sequence, while the x-axis represents the text sequence. The even-numbered labels represent the blank token in CTC. The color gradients represent the (log-linearly scaled) probability mass of each CTC state.

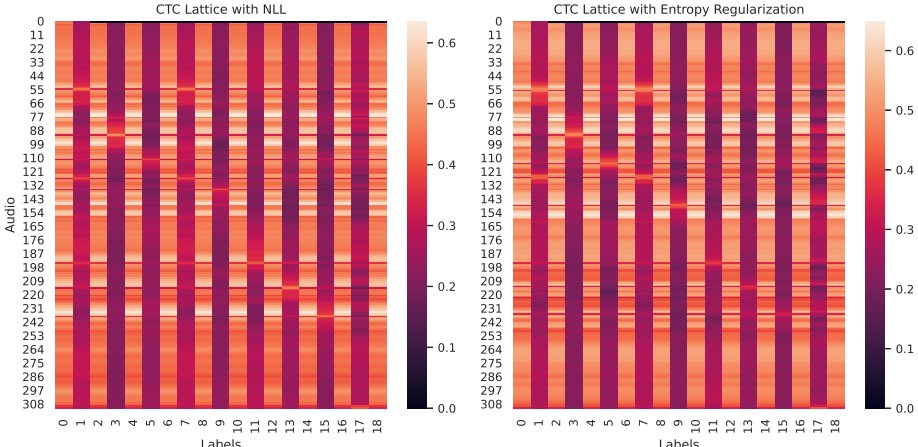

Figure 6: Visualization of CTC lattices similar to Fig. 5, but with the probability mass erroneously concentrated in wrong states in the lattice, for example (Audio = 55, Labels = 7).

