# OpenReview forum: "Revisiting the Entropy Semiring for Neural Speech Recognition"
_ICLR.cc/2023/Conference — ICLR 2023 poster_

### Official Review · Reviewer_dC8C · 2022-10-24

**Confidence:** 4
**Correctness:** 4
**Technical Novelty And Significance:** 4
**Empirical Novelty And Significance:** 4
**Recommendation:** 10

**Clarity, Quality, Novelty And Reproducibility:**

Clarity: Very clear, pleasure to read
Novelty: Novel, significant contribution
Reproducibility: empirical results should be reproducible

**Strength And Weaknesses:**

Pros:
* A novel regularizer and distillation objective based on entropy of alignment distribution achieving state of the art performance on Librispeech in streaming scenario.
* Open-source stable and parallel implementation for CTC and RNN-T

Cons:
N/A


**Summary Of The Paper:**

This paper proposes use of entropy of speech-text alignment distribution as a regularizer during model training or as an additional supervision signal in the model distillation process.  As a regularizer the alignment entropy attempts to prevent the alignment distribution that’s induced without any supervision from becoming too concentrated on some, possibly erroneous, alignments.  And during model distillation the alignment entropy is intended to provide additional supervision to the student model.  The key idea in the proposed approach is to utilize dynamic programming with entropy semi-ring to efficiently compute the alignment entropy.  Authors propose and open-source a numerically stable and parallel implementation of CTC and RNN-T within the semi-ring framework.

**Summary Of The Review:**

Paper makes a novel and significant contribution to enable use of alignment entropy as part of model training process.

---

> ### Author Response · Authors · 2022-11-17
> **Response**
>
> Thank you for your kind words and helpful comments about our manuscript.

---

### Official Review · Reviewer_NK1H · 2022-10-25

**Confidence:** 3
**Correctness:** 4
**Technical Novelty And Significance:** 3
**Empirical Novelty And Significance:** 3
**Recommendation:** 8

**Clarity, Quality, Novelty And Reproducibility:**

Clarity: Good. Clearly written except some typos.
Minor suggestions:
- $x_3 $ should be $ v_3$ in Example 2.10.
- Motvation -> Motivation in Section 5.1.

Quality: Good. Mathematical expressions are given correctly. Experiments show gains in both entropy regularization and in knowledge distillation experiments.
- In the regularization experiments, the batch size is 2048. How would this choice affect the regularization performance? Is the proposed model efficient memory-wise so that one can train with such a large batch size?
- A further analysis of the performance on sequences with different (short and long) lengths might have provided some further intuition for one of the motivations behind the proposed methods.

Novelty: Sufficiently novel. It revives the FST framework in the case of neural network training by building on the earlier studies on semi-rings.

Reproducibility: Even though some details such as the exact architecture of the RNNT model is missing (e.g. the structure of the predictor network), the paper is probably going to be accompanied by the open-source code (based on the claims in the text). Hence, it should become reproducible.

**Strength And Weaknesses:**

Strengths:
- The paper brings the mathematically backed up idea of semi-rings back to ASR such that they can be used in CTC or RNNT systems.
- The paper is clearly written.
- Experimental results show good WER performance in the Librispeech setting.

Weaknesses:
- In Table 3, does any of the papers utilize the LibriLight data? If yes, it might be better to denote that in the table. Current semiring knowledge distillation proposal might be benefiting from that external data and the comparison might not be fair enough.
- The appendix might be improved by giving some more details on the gradient back propagation over the semiring.

**Summary Of The Paper:**

This paper introduces a numerically stable and parallelizable version of the entropy semi-ring for ASR training. For numerical stability, the paper introduces log entropy semiring with $<\log p(e), \log(-p(e)\log p(e))> $. For the knowledge distillation, which is based on the hard and soft labels, the paper proposes one more semi-ring called log reverse-KL semiring by concatenating log and log entropy semi-rings. This way, the paper tries to incorporate alignment information into the distillation loss. Experiments show the effectiveness of the method on Librispeech in two settings: (1) adding entropy regularization to CTC/RNNT LSTM/Conformer combinations reduce the WER especially when the base model is not very strong. (2) In the RNN-T distillation experiments add Libri-Light as the semisupervised data which are pseudo labeled by a strong supervised model, then a teacher is learned using a combination of these LibriLight and LibriSpeech data sets. The student model is trained on LibriLight using the teacher. The results show that the proposed distillation approach performs better than hard and soft distillation alone. Comparison with streaming models also show the success of the proposed approach. The paper will also be accompanied with an open-source implementation of the proposed approach.

**Summary Of The Review:**

The paper brings the mathematically backed up idea of semi-rings back to ASR such that they can be used in CTC or RNNT systems. The paper provides both mathematical description and experimental evaluation of the proposed semirings. Experiments on entropy regularization and knowledge distillation show low enough WERs. The paper is well-written. Open source implementation will be provided and the paper should become reproducible. It is a good paper in general.

---

> ### Author Response · Authors · 2022-11-17
> **Response**
>
> Thank you for your kind words and helpful comments about our manuscript.
>
> > "In Table 3, does any of the papers utilize the LibriLight data? If yes, it might be better to denote that in the table"
>
> We have included this information in Table 3. We note that most prior work do not use the unlabeled Librilight data, and acknowledge the reviewer’s concern that the use of the additional unlabeled data makes a fair comparison difficult. However, we note the following points:
>
> 1) With the exception of Hwang et al., none of the cited prior work does distillation, for which using unlabeled data is a natural thing to do. Hwang et al. does do distillation and also uses Librilight unlabeled data, and we do report better results.
>
> 2) The comparison with Chiu et al. should be considered a fair comparison, since we use the same architecture and the same training data (Chiu et al. uses Librilight as well). In this case, our resultant model significantly outperforms Chiu et al., with a 29% improvement on Test-Clean and a 21% improvement on Test-Other.
>
> 3) Our work also does not use any future context at all in making predictions, which is a handicap compared to prior work that does use it. Even a small amount of look-ahead can improve accuracy substantially, with a 12% improvement for Zhang et al. and a 10% improvement for Moritz et al.
>
> > "The appendix might be improved by giving some more details on the gradient back propagation over the semiring."
>
> In [1], Eisner notes that instead of deriving the inside-outside algorithm by hand, which can be extremely tricky, we can get it "for free" in a simple way by relying on backpropagation and automatic differentiation. Similarly, our semiring implementation for CTC and RNN-T does not invoke any special code paths or customized over-rides for the backward pass. It simply defines a numerically stable forward pass, and relies on Tensorflow's automatic differentiation library to do the rest. We encourage the reader to lean into the "magic" of modern automatic differentiation, and not have to worry about what the semiring for the backward pass looks like.
>
> [1] Inside-outside and forward-backward algorithms are just backprop. Jason Eisner.
>
> > "Clearly written except some typos"
>
> Thanks for pointing out the typos. We have corrected them.
>
> > "In the regularization experiments, the batch size is 2048. How would this choice affect the regularization performance?"
>
> We did not investigate how batch size affects regularization performance, but we do not think that a large batch size is necessary in general.
>
> > "Is the proposed model efficient memory-wise so that one can train with such a large batch size?"
>
> The short answer is yes.
>
> Regarding memory efficiency, note that the overhead scales as a multiple of the size of the alignment lattice. Here’s a back-of-the-envelope estimate for the size of the memory overhead. For RNN-T, the shape of the table is [batch size, audio length, text length]. Say we have float32 tensors, batch size 16, audio length 1000, text length 1000, that amounts to 4 bytes * 16 * 1000 * 1000 * 2 * 2 = 256Mb. For regularization, the new table is twice as big, so a 256Mb overhead, and for distillation which is four times as big, a 768Mb overhead. For a 12Gb GPU, this amounts to only 2% and 6% of the available memory respectively.
>
> Note that batch size 16 refers to the per-device batch, and the overall batch size is much larger due to data parallelism, i.e. multiple devices are used. (For reference, our experiments in Section 5.1 use a per-device batch size of 32, and those in Section 5.2 use a per-device batch size of 8.) Our calculation involves multiplying by 2 twice at the end. The first doubling is for the two sets of logits: the logit to produce the text token and the logit to consume the acoustic token. The second doubling is for the forward pass and the backward pass.
>
> In general, the actual memory overhead is highly dependent on specific implementation details like the hardware, the parallelism strategy, gradient checkpointing, quantization, compiler and intermediate representations, etc. But in most cases, it should be a tiny fraction of the memory usage, and not something to be concerned with.
>
> > "A further analysis of the performance on sequences with different (short and long) lengths might have provided some further intuition"
>
> The focus of this paper was on a numerically stable implementation, which required the mathematical proposal of different variants of the entropy semiring as well as the difficult coding implementation of the semiring framework for CTC and RNN-T. We have observed that even on short sequences (< 100), a naive implementation of the entropy semiring will not result in successful training due to NaNs, but did not determine exactly when it breaks. We agree that further theoretical and empirical analysis of the optimality of alignments and how that varies with sequence length is an important research topic, and we leave it to future work.

---

### Official Review · Reviewer_3ceX · 2022-10-25

**Confidence:** 3
**Correctness:** 3
**Technical Novelty And Significance:** 3
**Empirical Novelty And Significance:** 3
**Recommendation:** 6

**Clarity, Quality, Novelty And Reproducibility:**

Overall, this paper is clearly written, with well structured sections on background, methodology, experiments etc. It proposes new approaches to improve neural speech recognition, and it is reproducible with open-source implementation.

**Strength And Weaknesses:**

Strength: The idea of regularization or distillation based on alignment entropy is technically sound. Overall, this paper is well structured, with detailed background introduction on semiring. Experiments are well-designed for validation purpose. Open-source contribution with this work makes it easier for speech community to experiment with and integrate this approach.

Weakness: I think experimental section could be made stronger with more details added to explain design and results. See questions in the summary section below.

**Summary Of The Paper:**

In this paper authors propose to leverage entropy semiring and alignment entropy to improve the performance of neural speech recognition, via regularization or distillation. Experimental results show its effectiveness. There are also open-source contributions based on this work.

**Summary Of The Review:**

The methodology presented is technical sound, with convincing experimental design and results. Open-source implementation makes it easier for speech community to leverage this work. I think the experimental section could be made stronger by adding more details to justify design and results, including:

1. Adding std to those results in Table 1-3 to show if improvements are statistically significant.
2. How to determine model hyperparameters, e.g. those listed in the "Experimental Setup" paragraphs in Sec 5.1 & 5.2.
3. As shown in Table 3, prior work has different model sizes. Do authors also experiment with smaller models (e.g. similar as 30M or 80M #Params)?

---

> ### Author Response · Authors · 2022-11-17
> **Response**
>
> Thank you for your kind words and helpful comments about our manuscript.
>
> > "I think experimental section could be made stronger with more details added to explain design and results"
>
> We have included additional details in the description of the experimental setup. Please let us know if there are specific details we have omitted that you would like us to include. Unfortunately, due to space constraints, our descriptions might not be completely self-contained for readers less familiar with the literature. Where appropriate, we have cited specific literature for the reader to learn more. For example, we report in Section 5.2 that we use the exact same causal Conformer architecture as Chiu et al., without describing how their random projection quantizer works. We believe that the interested reader will be able to learn more details about it by reading Chiu et al.
>
> > "Adding std to those results in Table 1-3 to show if improvements are statistically significant"
>
> We have included 95% WER confidence intervals in Tables 1 and 2. Unfortunately, none of the prior work in Table 3 report confidence intervals or do significance testing.
>
> > "How to determine model hyperparameters"
>
> We mostly followed the same architectural hyper-parameters that were used in prior work. For example, the Conformer hyper-parameters used in Section 5.1 stick to the same hyper-parameters used in Gulati et al., and Section 5.2 adopts the exact same architectural hyperparameters as Chiu et al. (2022). Optimization hyper-parameters generally follow Vaswani et al. (2017) in using Adam, and a learning rate schedule that does a linear warm-up followed by annealing. The learning rate and batch size were chosen to enable speedy and successful training on our infrastructure.
>
> > "Do authors also experiment with smaller models (e.g. similar as 30M or 80M #Params)?"
>
> We conducted an additional experiment by shrinking the size of the student model in Section 5.2 to a 30M model. This is similar to the Conformer M model (30M) in Gulati et al. or the Conformer model (30M) by Yu et al. (2020) cited in Table 3.
>
> We observe the following results:
>
> Dev-Clean: 3.1 +- 0.2,
>
> Dev-Other: 9.5 +- 0.4,
>
> Test-Clean: 3.2 +- 0.2,
>
> Test-Other: 9.5 +- 0.4.
>
> These results compare favorably with Yu et al. (2020) who report:
>
> Test-Clean: 3.7,
>
> Test-Other: 9.2,
>
> with significantly better performance on Test-Clean (14% improvement), and slightly worse performance on Test-Other (3% degradation).

---

### Official Review · Reviewer_jsaa · 2022-10-28

**Confidence:** 5
**Correctness:** 3
**Technical Novelty And Significance:** 3
**Empirical Novelty And Significance:** 4
**Recommendation:** 10

**Clarity, Quality, Novelty And Reproducibility:**

Clarity

- the paper is well written.
- semiring explanations are easy to understand
- some parts (e.g., Example 2.8) are difficult to follow, but later I found the solution in the appendix. It is better to add some pointers to the appendix.
- the motivation of corresponding experiments is also evident.

Quality

- The elegant formulation and reasonable experimental results make this paper's quality very high.

Novelty

- Although entropy semiring itself is not novel, I think the authors' contributions to the backprobable algorithm have enough novelty.

Reproducibility

- The method will be open-sourced. Together with the public Librispeech benchmarks, the paper has high reproducibility


**Strength And Weaknesses:**

Strength

- clear descriptions of each semiring operations
- novel entropy semiring formulation by considering the backpropagation capability and efficiency.
- reasonable experimental results showing the mitigation of overconfidence and the knowledge distillation
- archives the state-of-the-art performance in the Librispeech streaming setup

Weaknesses

- it could have more discussions about other applications than ASR (although they are briefly mentioned in the conclusion section).
- lack of the analysis
- there are some duplicated descriptions (e.g., the first paragraph in Section 3 and the first paragraph in Section 3 are highly overlapped with Section 1). These parts can be improved.

**Summary Of The Paper:**

This paper proposes a new speech recognition formulation based on the entropy semiring, which is used as an entropy regularization and knowledge distillation. First, the paper motivates hard alignment-based speech recognition issues based on overconfidence and the solution based on entropy regularization. Then, the paper provides an elegant and easy-to-understand formulation of the semiring from the basic one to their novel entropy semiring by caring for the backpropagation. The experiments also show the expected result of using entropy regularization and knowledge distillation experiments for the well-known Librispeech benchmarks with state-of-the-art performance in a streaming setup.



**Summary Of The Review:**

The paper's novelty, clearness, experimental effectiveness, and reproducibility significantly contribute to the machine-learning community. I strongly recommend this paper be accepted.

Other suggestions
- It would be better to provide some more concrete examples for the semiring explanation (e.g., Example 2.6 corresponds to the forward computation, right? We can provide such an example)
- Can we visualize whether the overconfidence is mitigated or not in Section 5.1? I think such analysis (visualization) strengthens the claim of this paper.
- Section 5.1: It might significantly improve the performance in the out-of-domain conditions (e.g., noisy speech recognition, multilingual speech recognition). I would recommend the authors apply such tasks to prove further the effectiveness of mitigating overconfidence by the proposed method.

---

> ### Author Response · Authors · 2022-11-17
> **Response**
>
> Thank you for your kind words and helpful comments about our manuscript.
>
> > "it could have more discussions about other applications than ASR"
>
> In the Conclusion section, we have expanded our discussion of potential applications involving multiple modalities or domains, as well as suggested additional directions for future work. In the Related Work section, we have expanded our discussion about the use of semirings in relevant machine learning literature.
>
> > "there are some duplicated descriptions (e.g., the first paragraph in Section 3 and the first paragraph in Section 3 are highly overlapped with Section 1)"
>
> We have shortened the first paragraph of Section 3 and made our writing more concise.
>
> > "It would be better to provide some more concrete examples for the semiring explanation"
>
> We have included extended step-by-step derivations of Examples 2.8 and 2.10 in the style of Example 2.6 to the Appendix, and included pointers to them in the main text. This should make it easier for the reader to understand them.
>
> > "Can we visualize whether the overconfidence is mitigated or not in Section 5.1?"
>
> We agree that some kind of visualization of the alignments will be helpful. It is not clear what the best way of doing this is, but we think perhaps a heat map of some sort will be helpful. We will spend more time thinking about this, and include it in the camera-ready version of our paper should it be accepted.
>
> > "It might significantly improve the performance in the out-of-domain conditions (e.g., noisy speech recognition, multilingual speech recognition)."
>
> We agree with the reviewer that more work needs to be done to investigate the effectiveness of our proposed approach in different domains like noisy speech recognition and multilingual speech recognition. We hope that our open-source contribution will make it easy for future work to conduct such investigations.

---

> > ### Comment · Reviewer_jsaa · 2022-11-29
> > **Thanks for your response.**
> >
> > Thanks for your response.
> > I'm looking forward to updates in the analysis part and the effectiveness in different domains.

---

> > > ### Author Response · Authors · 2022-11-30
> > > **Response**
> > >
> > > We have now completed our visualization and done a further qualitative analysis on the effect of alignment entropy regularization. Interestingly, it seems like its main effect is to encourage more local exploration of alignment paths around the "peaks" in the lattice, rather than to avoid them altogether. This results in flatter but denser local optima, which helps on average. However, on examples where the entropy regularized model does worse than the NLL model, we see that it is actually more over-confident and concentrates even more in the local vicinity of the misplaced peaks. This shows that even though alignment entropy regularization is helpful on average for reducing model over-confidence, it can actually backfire in some instances. This finding suggests important directions for future work (potentially a new semiring) that encourages more global, rather than / in addition to local, exploration of alignment paths that are much further away from existing local optima.
> > >
> > > We thank the reviewer again for their suggestion to study empirical visualizations in the alignment space. This has provided stronger intuition for the effects of the regularization mechanism and uncovered potential directions for future work.

---

### Decision · Program_Chairs · 2023-01-20

**Decision:**

Accept: poster

**Justification For Why Not Higher Score:**

The novelty technically is not as significant as the reviewers have commented, and the authors implicitly acknowledge that by having the word "revisit" in the title. Eisner and Mohri have both independently propose semirings with tuples that can achieve interesting computations, such as for EM and for k-best decoding. The approach saves the effort of implementing dynamic programming, but does not actually save any computation.

**Justification For Why Not Lower Score:**

The formalism of the paper has great merit, and can potentially spawn the design other training losses.

**Metareview: Summary, Strengths And Weaknesses:**

The paper uses entropy semiring to implement entropy regularization and sequence knowledge distillation. The framework is mathematically clean and elegant. Experiments on automatic speech recognition show small but consistent improvements.

One reviewer has concern about whether the amount of improvement is significant. The improvement depends on entropy regularization and knowledge distillation, not the semiring itself. The experiments are proof of concepts, so I am not too concerned about the small amounts of improvement.

All reviewers unanimously praise the writing the elegance of the framework.

**Note From Pc:**

if the above contains the word "oral" or "spotlight" please see: "oral" presentation means -> notable-top-5% and "spotlight" means -> notable-top-25%. As stated in our emails, we are disassociating presentation type from AC recommendations